# On Representing Electronic Wave Functions with Sign Equivariant Neural Networks

**Nicholas Gao, Stephan Günnemann**
{n.gao,s.guennemann}@tum.de
Department of Computer Science & Munich Data Science Institute
Technical University of Munich

## Abstract

Recent neural networks demonstrated impressively accurate approximations of electronic ground-state wave functions. Such neural networks typically consist of a permutation-equivariant neural network followed by a permutation-antisymmetric operation to enforce the electronic exchange symmetry. While accurate, such neural networks are computationally expensive. In this work, we explore the flipped approach, where we first compute antisymmetric quantities based on the electronic coordinates and then apply sign equivariant neural networks to preserve the antisymmetry. While this approach promises acceleration thanks to the lower-dimensional representation, we demonstrate that it reduces to a Jastrow factor, a commonly used permutation-invariant multiplicative factor in the wave function. Our empirical results support this further, finding little to no improvements over baselines. We conclude with neither theoretical nor empirical advantages of sign equivariant functions for representing electronic wave functions within the evaluation of this work.

## 1 Introduction

The stationary Schrödinger equation (Schrödinger, 1926) at the heart of quantum chemistry is a partial differential equation in $3N$ dimensions, where $N$ is the number of electrons in the system:

$$\boldsymbol{H}\psi = E\psi \tag{1}$$

where $\psi : \mathbb{R}^{N \times 3} \to \mathbb{R}$ is the wave function, $E$ the energy and $\boldsymbol{H}$ the Hamiltonian of the system, see Appendix F. Electronic wave functions must obey the exchange antisymmetry $\psi(..., r_i, ..., r_j, ...) = -\psi(..., r_j, ..., r_i, ...)$. Typically, one enforces this by using a so-called Slater determinant

$$\psi(\boldsymbol{r}) = \det [\phi_j(r_i)]_{i,j \in \{0,...,N\}} = \det \Phi(\boldsymbol{r}). \tag{2}$$

In practice, one often uses linear combinations of determinants $\psi(\boldsymbol{r}) = \sum_{k=1}^{K} w_k \det \Phi_k(\boldsymbol{r})$. Further, note that the antisymmetric property is preserved by multiplying the antisymmetric function with a symmetric function $J : \mathbb{R}^{N \times 3} \to \mathbb{R}$, leading to the so-called Slater-Jastrow wave function

$$\psi(\boldsymbol{r}) = J(\boldsymbol{r}) \sum_{k=1}^{K} w_k \det \Phi_k(\boldsymbol{r}). \tag{3}$$

Recent works improved these Ansätze by replacing the so-called orbital functions $\phi_i$ in Equation 2 by neural networks (Hermann et al., 2023). While this has shown remarkably accurate approximations of the electronic wave function of various molecules, it comes at a significant computational cost due to explicit electron-electron interactions. In classical quantum chemistry, one avoids this high cost by using easily integrable Gaussian basis functions to construct the orbital functions $\phi_i$. Unfortunately, due to the simple structure of the orbital functions, one may typically require thousands to millions of determinants to capture electronic correlations correctly (Gao et al., 2024).

In this work, we aim to reduce the classical large number of determinants via neural networks. But, instead of defining the orbital functions via neural networks, we explore non-linear combinations of determinants, i.e., we are reformulating the classic Slater-Jastrow wave function to

$$\psi(\boldsymbol{r}) = f\left([\det \Phi_k(\boldsymbol{r})]_{k=1}^{K}, J(\boldsymbol{r})\right), \tag{4}$$

where $f(\boldsymbol{x}, y) = y \sum_{k=1}^{K} w_k x_k$ recovers the classical case. To accomplish this, we define a set of sign equivariant operations to preserve the antisymmetry of the determinants in the final output. Our theoretical analysis finds that such an algebraic construction is identical to a Jastrow factor and, thus, cannot shrink the zero set of the wave function. We support these findings via empirical results.

## 2 RELATED WORK

Solving the stationary Schrödinger equation accurately for many systems opens the path to fast and accurate machine-learned force fields (Schütt et al., 2018; Kosmala et al., 2023; Wollschläger et al., 2023). The machine learning approach to this has been, since the first work by Lou et al. (2023), to parameterize the orbitals from from Equation 2 with neural networks (Hermann et al., 2023; Zhang et al., 2023). While subsequent works tweak architectures (Gerard et al., 2022; von Glehn et al., 2023), explore new applications (Cassella et al., 2023; Kim et al., 2023; Lou et al., 2023; Wilson et al., 2022; Pescia et al., 2023), compute excited states (Pfau et al., 2023; Entwistle et al., 2022), generalize across molecules (Scherbela et al., 2022; 2024; 2023; Gao & Günnemann, 2022; 2023b;a), or explore the use of Diffusion Monte Carlo (Wilson et al., 2021; Ren et al., 2023), the underlying structure of the wave function remained the same. This work explores a different approach by parametrizing the wave function via non-linear combinations of determinants.

## 3 SIGN EQUIVARIANT FUNCTIONS

Throughout this work, we will mainly refer to two different symmetries. Firstly, the fermionic antisymmetry to permutations, i.e., $\psi(\pi(\boldsymbol{r})) = \text{sign}(\pi)\psi(\boldsymbol{r})$, and, secondly, to odd function, more formally functions that are equivariant to the cyclic group $C_2$, i.e., $f(-x) = -f(x)$. The framework of equivariance allows us to define these symmetries more generally.

**Definition 1.** *A function $f : \mathcal{X} \rightarrow \mathcal{Y}$ on real vector spaces $\mathcal{X}, \mathcal{Y}$ is said to be equivariant under group $\mathbb{G}$ iff $\forall g \in \mathbb{G}$ $f(G_g^{\mathcal{X}} x) = G_g^{\mathcal{Y}} f(x)$ where $G_g^{\mathcal{X}}, G_g^{\mathcal{Y}}$ are the group representations of $g$ acting on the vector spaces $\mathcal{X}$ and $\mathcal{Y}$, respectively.*

As a special case of equivariance, one can define invariance where the result of a function does not change under group actions, i.e., $G_g^{\mathcal{Y}}$ is the identity for all $g \in \mathbb{G}$. Given these definitions, we can now concretely describe antisymmetric and odd functions. Further, to aid later discussion, we also introduce symmetric and even functions as counterparts.

**Definition 2.** *A function $f : \mathcal{X} \rightarrow \mathcal{Y}$ is* antisymmetric *iff $f$ is equivariant under the symmetric group $S_n$ and the group acts on $\mathcal{Y}$ as $\boldsymbol{y} \mapsto \text{sign}(\pi)\boldsymbol{y}, \forall \pi \in S_n, \boldsymbol{y} \in \mathcal{Y}$.*

**Definition 3.** *A function $f : \mathcal{X} \rightarrow \mathcal{Y}$ is* symmetric *iff $f$ is invariant under the symmetric group $S_n$.*

**Definition 4.** *A function $f : \mathcal{X} \rightarrow \mathcal{Y}$ is* odd *iff $f$ is equivariant under the cyclic group $C_2 = \{-1, 1\}$ and the group acts on $\mathcal{Y}$ as $\boldsymbol{y} \mapsto g\boldsymbol{y}, \forall g \in C_2, \boldsymbol{y} \in \mathcal{Y}$.*

**Definition 5.** *A function $f : \mathcal{X} \rightarrow \mathcal{Y}$ is* even *iff $f$ is invariant under the cyclic group $C_2$.*

**Implicit Odd Functions** A function $f$ acting on the vector of determinants $\boldsymbol{x} = [\det \Phi_k(\boldsymbol{r})]_{k=1}^{K}$ must be odd, i.e., $f(-x) = -f(x)$, to preserve the fermionic antisymmetry. Let $f_1$ and $f_2$ be odd functions and $g$ an even function. The following functions are odd: (1) multiplication with a constant $\alpha f(x)$, (2) addition $f_1(x) + f_2(x)$, (3) element-wise odd functions, e.g., $f(x) = [\tanh(x_i)]_{k=1}^{K}$, (4) multiplication with an even function $g$, i.e., $f(x)g(x)$, and (5) chaining $f_2(f_1(x))$. With (5), we can compose complex functions by chaining simpler ones. Combining (1) and (2) yields linear combinations, i.e., we can construct linear layers without bias terms. We construct neural networks $f^{(T)}$ by combining this with non-linear activation functions (3):

$$f^{(t+1)}(\boldsymbol{x}^{(t)}) = \tanh\left(\boldsymbol{x}^{(t)} \boldsymbol{W}^{(t)}\right) * J^{(t)} \tag{5}$$

where $J^{(t)}$ is obtained from a Jastrow factor, i.e., a symmetric function of the electronic coordinates.

**Explicit Odd Functions.** Alternatively, given an arbitrary function $g : \mathcal{X} \rightarrow \mathcal{Y}$, one can construct an odd function $f : \mathcal{X} \rightarrow \mathcal{Y}$ via $f(x) = g(x) - g(-x)$. Inversely, as proven in Appendix A, every odd function $f$ can be expressed by a non-odd function $g$:

**Theorem 1.** *Any odd function $f : \mathcal{X} \rightarrow \mathcal{Y}$ on real vector spaces $\mathcal{X}, \mathcal{Y}$ can be represented as $f(\boldsymbol{x}) = g(\boldsymbol{x}) - g(-\boldsymbol{x})$ where $g : \mathcal{X} \rightarrow \mathcal{Y}, \boldsymbol{x} \in \mathcal{X}$.*

In our experiments, we implement $g$ via an MLP with Jastrow factors $J^{(0)}$ and $J^{(T)}$:

$$f(\boldsymbol{x}) = \left(\text{MLP}\left(\boldsymbol{x} * J^{(0)}\right) - \text{MLP}\left(-\boldsymbol{x} * J^{(0)}\right)\right) * J^{(T)}. \tag{6}$$

**Linear-logarithmic Domain.** One may notice that the distribution of $\psi$ for samples drawn from $\psi^2$ varies by several orders of magnitude. We illustrate this in Appendix C for LiH. While well separated in the logarithmic domain, the data is crammed into a small region when viewed in the linear domain. Unfortunately, we cannot directly work in the log-domain as it prohibits any intermediate values from being zero. We address this by working in a domain that resembles a linear relationship around zero and a logarithmic relationship far from zero. We define the linlog transformation as

$$\text{linlog}_\alpha(x) = \text{sign}(x) \log(|x| e^\alpha + 1) \tag{7}$$

where $\alpha$ controls a shift of the data towards the linear or logarithmic region. We depict this function and discuss its stable implementation in Appendix D.

## 4 THEORETICAL RESULTS

**Equivalence to Jastrow Factor.** Before empirically analyzing the sign equivariant functions, we point out an equivalence between odd functions and a Jastrow factor in the context of electronic wave functions. Interestingly, one can show that all functions of the form in Equation 4 can be represented by functions in the classical Slater-Jastrow form in Equation 3.

**Theorem 2.** *Let $\mathcal{R}, \mathcal{X}, \mathcal{Y}, \mathcal{Z}$ be real vector spaces, $\phi : \mathcal{R} \rightarrow \mathcal{X}$ an antisymmetric function, $J : \mathcal{R} \rightarrow \mathcal{Y}$ a symmetric function, and $f : \mathcal{X} \times \mathcal{Y} \rightarrow \mathcal{Z}$ an in the first argument odd function, i.e., $f(-\boldsymbol{x}, \boldsymbol{y}) = -f(\boldsymbol{x}, \boldsymbol{y})$. Any antisymmetric function $\psi : \mathcal{R} \rightarrow \mathcal{Z}; \psi(\boldsymbol{r}) = f(\phi(\boldsymbol{r}), J(\boldsymbol{r}))$ can be expressed a.e. as $\psi(\boldsymbol{r}) = (\phi(\boldsymbol{r})^T \boldsymbol{x})\hat{J}(\boldsymbol{r})$ with $\boldsymbol{x} \in \mathcal{X} \setminus \{\boldsymbol{0}\}$ and a symmetric function $\hat{J} : \mathcal{R} \rightarrow \mathcal{Z}$.*

We prove this theorem in Appendix B. In the context of quantum chemistry, this result implies that a linear combination and a non-positive-constrained Jastrow factor can equally represent any odd non-linear combination of the antisymmetric function. While this result guarantees the existence of such a Jastrow factor, it provides no statement about the ease of finding such a solution.

**Cusp Conditions.** Thanks to Kato (1957)'s theorem, we know that the solutions to Equation 1 must fulfill the cusp condition $\lim_{\boldsymbol{r} \rightarrow R_m} -\frac{1}{\psi(\boldsymbol{r})}\frac{\partial \psi(\boldsymbol{r})}{\partial \boldsymbol{r}} = Z_m$ where $R_m, Z_m$ are the position and charge of the $m$-th nucleus, respectively. However, one cannot fulfill the cusp conditions if one chooses Gaussian-type orbitals as basis functions $\phi_i$ in Equation 2. In fact, all extrema of the original wave function are preserved. One can verify this via the chain rule on $\psi(\boldsymbol{r}) = f([\det \Phi_k(\boldsymbol{r})]_{k=1}^K)$:

$$\frac{\partial \psi(\boldsymbol{r})}{\partial \boldsymbol{r}} = \frac{\partial \psi(\boldsymbol{r})}{\partial [\det \Phi_k(\boldsymbol{r})]_{k=1}^K} \frac{\partial [\det \Phi_k(\boldsymbol{r})]_{k=1}^K}{\partial \boldsymbol{r}}. \tag{8}$$

Thus, anytime the orbital derivatives are 0, the derivatives of the final wave function will be 0. As the derivatives of Gaussian-type orbitals are zero at the nuclei, the cusp conditions cannot be fulfilled.

## 5 EMPIRICAL RESULTS

While Theorem 2 guarantees the existence of equivalent Jastrows, it does not provide insights into finding one. Thus, we experimentally evaluate the impact of odd functions on the energy of wave functions. We test odd functions on four different systems, LiH, Li2, and two states of N2, as detailed in Appendix E. We use two different approaches to get the initial antisymmetric input for the odd functions: (1) CASSCF wave functions, a series of Slater determinants from standard quantum chemistry (Szabo & Ostlund, 2012), and (2) FermiNet, a neural network wave functions (Pfau et al., 2020). For optimization, we use the variational Monte Carlo (VMC) framework, which we highlight in Appendix F, with K-FAC (Martens & Grosse, 2015). Ablations with Prodigy (Mishchenko & Defazio, 2023) can be found in Appendix H and the exact hyperparameters in Appendix G.

**CASSCF.** We first use the cheaper CASSCF wave functions to test the hyperparameter $\alpha \in \{-2, 0, 2\}$'s impact from Equation 7 on the final energy. Additionally, we repeat the experiment in the linear and linlog domain. Further, we compare a classical linear readout, implicit odd functions from Equation 5, and the explicit odd functions from Equation 6. The final energies are plotted

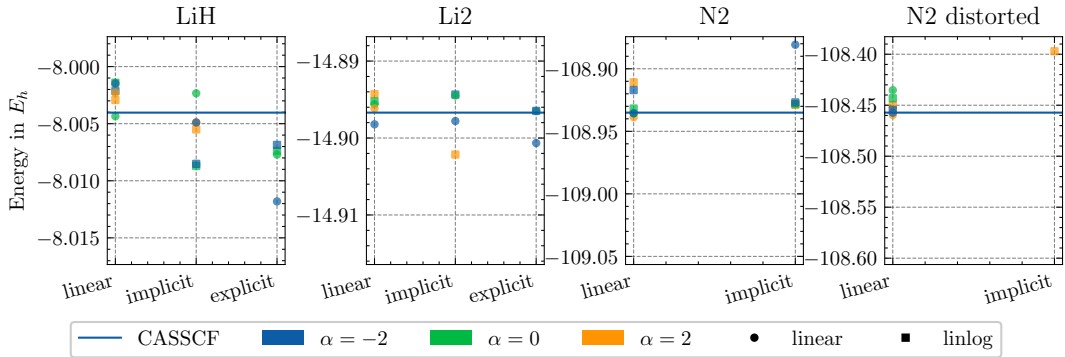

Figure 1: CASSCF for different values of $\alpha$ without symmetric functions.

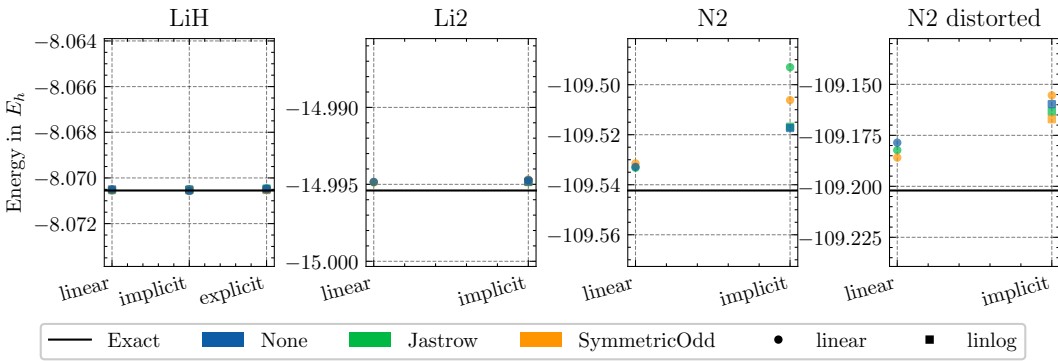

Figure 2: FermiNet with different choices of symmetric functions and $\alpha = -2$.

in Figure 1. On small structures like LiH and Li2, the explicit odd functions improve energies by up to $8\,\mathrm{m}E_\mathrm{h}$ and $4\,\mathrm{m}E_\mathrm{h}$, respectively. Unfortunately, this advantage does not carry over to larger, more challenging molecules like the nitrogen dimer. We observe significant numerical instabilities with implicit or explicit odd functions in the optimization process; we explore this further in Appendix H.

**FermiNet.** For FermiNet we picked the best performing $\alpha$ from the CASSCF experiments, i.e., $\alpha = -2$. We evaluate the energy change by including a Jastrow factor. We consider three types of Jastrows, (1) none, (2) a traditional Jastrow, Equation 3, (Jastrow), and (3) a Jastrow included the odd function, Equations 5 and 6, (SymmetricOdd). Appendix G details the exact Jastrow function. In contrast to CASSCF wave functions, FermiNet can closely recover the ground state thanks to the explicit inclusion of electron correlation in the orbitals. We plot the final energies in Figure 2. However, neither odd function positively impacts the optimization but worsens energies. Adding a Jastrow factor leads to improvements in the more challenging nitrogen dimer.

## 6 CONCLUSION

Moving neural network complexity from the orbital functions to a non-linear combination of simpler basis functions promises faster quantum chemistry methods than deep neural networks applied to electronic coordinates. To design such functions, we present two approaches: implicit constructions by combining simple odd functions and explicitly odd functions where one enforces the antisymmetry at the end. While our experimental results show that such odd functions combined with classical determinants can yield better results on small structures, optimizing such odd functions proves difficult. At larger system sizes, we frequently observe numerical instabilities and degraded performance. In combination with neural network wave functions, we found such odd functions to worsen energies. From a theoretical point of view, we have shown the limitations of odd functions. Specifically, they correspond to a subset of the set of functions obtainable via Jastrow factors. Given the theoretical and empirical results, we find little evidence for computational or accuracy advantages of non-linear combinations of Slater determinants for machine-learning quantum chemistry.

ACKNOWLEDGEMENTS

We thank Arthur Kosmala for his invaluable feedback on the manuscript, and Jan Schuchardt for providing additional financial support during the writing stage. Large parts of this work were done during Nicholas Gao's internship at Microsoft Research. Funded by the Federal Ministry of Education and Research (BMBF) and the Free State of Bavaria under the Excellence Strategy of the Federal Government and the Länder.

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

## A   PROOF OF THEOREM 1

*Proof of Theorem 1.* Define

$$g(\boldsymbol{x}) = \begin{cases} f(\boldsymbol{x}) & \text{, if } \boldsymbol{x}^T \boldsymbol{v} > 0, \\ \boldsymbol{0} & \text{, else,} \end{cases} \tag{9}$$

for some arbitrary non-zero vector $v \in \mathcal{X}_{/\boldsymbol{0}}$. This construction yields

$$g(\boldsymbol{x}) - g(-\boldsymbol{x}) = \begin{cases} f(\boldsymbol{x}) & \text{, if } \boldsymbol{x}^T \boldsymbol{v} > 0, \\ -f(-\boldsymbol{x}) & \text{, else} \end{cases} \tag{10}$$

$$\stackrel{f \text{ odd}}{=} f(\boldsymbol{x}). \tag{11}$$

$\square$

## B   PROOF OF THEOREM 2

**Lemma 1.** *Every odd function $f : \mathcal{X} \to \mathcal{Y}$ on real vector spaces $\mathcal{X}, \mathcal{Y}$ can be represented almost everywhere as a product of a linear combination $\boldsymbol{x}^T \boldsymbol{v}, \boldsymbol{v} \in \mathcal{X} \setminus \{\boldsymbol{0}\}$ and an even function $h : \mathcal{X} \to \mathcal{Y}, h(\boldsymbol{x}) = \frac{f(\boldsymbol{x})}{\boldsymbol{x}^T \boldsymbol{v}}$, i.e.,*

$$f(\boldsymbol{x}) = \boldsymbol{x}^T \boldsymbol{v} h(\boldsymbol{x}). \tag{12}$$

*Proof of Lemma 1.* We start with proving that $h(\boldsymbol{x}) = \frac{f(\boldsymbol{x})}{\boldsymbol{x}^T \boldsymbol{v}}$ for some $\boldsymbol{v} \in \mathcal{X}/\{\boldsymbol{0}\}$ is an even function

$$h(-\boldsymbol{x}) = \frac{f(-\boldsymbol{x})}{-\boldsymbol{x}^T \boldsymbol{v}} = \frac{-f(\boldsymbol{x})}{-\boldsymbol{x}^T \boldsymbol{v}} = \frac{f(\boldsymbol{x})}{\boldsymbol{x}^T \boldsymbol{v}} = h(\boldsymbol{x}). \tag{13}$$

Plugging $h$ into Equation 12 yields

$$\boldsymbol{x}^T \boldsymbol{v} h(\boldsymbol{x}) = \boldsymbol{x}^T \boldsymbol{v} \frac{f(\boldsymbol{x})}{\boldsymbol{x}^T \boldsymbol{v}} = f(\boldsymbol{x}). \tag{14}$$

$\square$

*Proof of Theorem 2.* Directly applying Lemma 1, we get

$$f(\phi(\boldsymbol{r}), J(\boldsymbol{r})) = (\phi(\boldsymbol{r})^T \boldsymbol{v}) \frac{f\left(\phi(\boldsymbol{r}), J(\boldsymbol{r})\right)}{\phi(\boldsymbol{r})^T \boldsymbol{v}}. \tag{15}$$

If we set $\hat{J}(\boldsymbol{r}) = \frac{f(\phi(\boldsymbol{r}), J(\boldsymbol{r}))}{\phi(\boldsymbol{r})^T \boldsymbol{v}}$, it remains to show that $\hat{J}$ is symmetric:

$$\hat{J}(\pi(\boldsymbol{r})) = \frac{f\left(\phi(\pi(\boldsymbol{r})), J(\pi(\boldsymbol{r}))\right)}{\phi(\pi(\boldsymbol{r}))^T \boldsymbol{v}} \tag{16}$$

$$= \frac{f\left(\text{sign}(\pi)\phi(\boldsymbol{r}), J(\boldsymbol{r})\right)}{\text{sign}(\pi)\phi(\boldsymbol{r})^T \boldsymbol{v}} \qquad \text{def. } \phi, J \tag{17}$$

$$= \frac{\text{sign}(\pi)f\left(\phi(\boldsymbol{r}), J(\boldsymbol{r})\right)}{\text{sign}(\pi)\phi(\boldsymbol{r})^T \boldsymbol{v}} \qquad \text{def. } f \tag{18}$$

$$= \hat{J}(r). \tag{19}$$

$\square$

## C   DISTRIBUTION OF WAVE FUNCTION AMPLITUDES

In Figure 3, we plot the distribution of wave function amplitudes for LiH in the logarithmic, linear, and linlog domains. We drew samples from $\psi^2$ and plotted the distribution of the amplitudes $\psi$. While the data looks well distributed in the logarithmic domain, it is crammed into a small region in the linear domain. Our linlog transformation from Equation 7 addresses this issue by viewing part of the data in the linear domain and part in the logarithmic domain. The scaling factor $\alpha$ controls the shift between the two domains. It plays a crucial role as a hyperparameter in our experiments. A small $\alpha$ leads to a spiky distribution like in the linear domain, and a large $\alpha$ leads to a flat distribution.

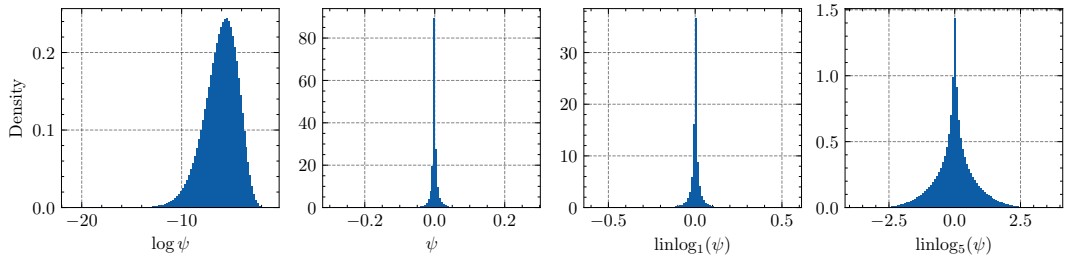

Figure 3: Distribution of wave function amplitudes for LiH in logarithmic (left), linear (center left), and lin-log (right figures) domain. Magnitudes vary between $10^{-2}$ and $10^{-20}$.

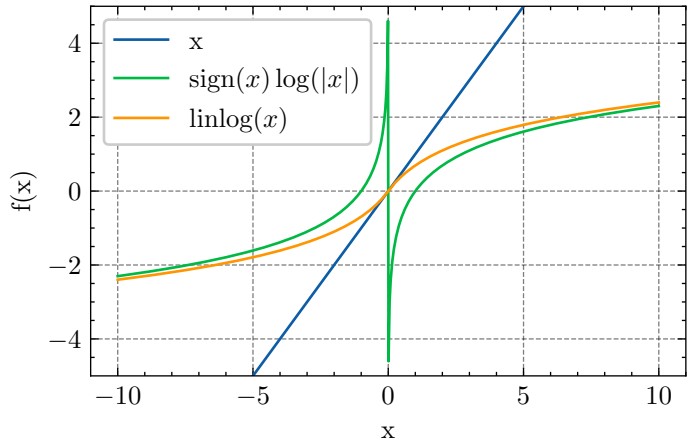

Figure 4: Illustration of different domains relative to linear data.

## D    LinLog Transformation

Our proposed linlog transformation from Equation 7 is $C^{\infty}$ smooth and can be numerically stably implemented to transform from and to the logarithmic domain directly via

$$\text{linlog}_{\alpha}(\text{sign}(x), \log(|x|)) = \text{sign}(x)\text{softplus}(\log(|x|) + \alpha). \tag{20}$$

To avoid the spiky distribution one typically observed in the linear domain, we initialize $\alpha = \alpha_{\text{init}} + \text{median}_i\{\max_k\{\det \Phi_k(\boldsymbol{r}_i)\}\}$ via a hyperparameter $\alpha_{\text{init}}$ and the median of a batch of electronic configurations sampled from $\psi^2$ with a linear readout instead of the learnable odd function. We plot the transformation relative to the linear domain in Figure 4.

## E    Used structures

We use four different systems for our empirical analysis: LiH, Li2, N2, and a distorted N2. For LiH and Li2, we pick the equilibrium structures from Pfau et al. (2020). For N2, we use the equilibrium structure from Pfau et al. (2020) as well a highly distorted structure where the error of FermiNet from Pfau et al. (2020) peaks. All structures are listed in Table 1. As active space for the CASSCF calculations, we pick all valence orbitals and electrons.

| Name | LiH | Li2 | N2 | N2 distorted |
|---|---|---|---|---|
| Distance ($a_0$) | 3.015 | 5.051 | 2.068 | 4.0 |
| Active space (Orbitals, Electrons) | (6, 2) | (6, 2) | (6, 10) | (6, 10) |

Table 1: Systems evaluated in this work.

## F   Variational Monte Carlo

Variational Monte Carlo (VMC) is a method to approximate solutions to the Schrödinger equation from Equation 1. In this work, we are interested in molecular systems where the Hamiltonian takes the following form in the Born-Oppenheimer approximation:

$$\boldsymbol{H} = -\sum_{i=1}^{N} \frac{1}{2} \frac{\partial^2}{\partial r_i^2} - \sum_{i=1}^{N} \sum_{m=1}^{M} \frac{Z_m}{\|r_i - R_m\|} + \sum_{i=1}^{N} \sum_{j=i+1}^{N} \frac{1}{\|r_i - r_j\|} + \sum_{m=1}^{M} \sum_{n=i+1}^{M} \frac{Z_m Z_n}{\|R_m - R_n\|}$$
(21)

with $r_i$ being the position of the $i$-th electron, $R_m$ the position of the $m$-th nucleus, and $Z_m$ the charge of the $m$-th nucleus. In linear algebra, Equation 1 is an eigenvalue problem, i.e., we are looking for the lowest eigenvalue $E_0$ and corresponding eigenvector $\psi_0$. To accomplish this, we use the variational principle, i.e., we approximate the ground state by a trial wave function $\psi_T$ and minimize the energy expectation value. The variational principle (Szabo & Ostlund, 2012) states that the energy expectation value of any trial wave function $\psi_\theta$ is an upper bound to the ground state energy $E_0$:

$$E_0 \leq \frac{\int \psi_\theta(\boldsymbol{r}) \boldsymbol{H} \psi_\theta(\boldsymbol{r}) \, \mathrm{d}\, \boldsymbol{r}}{\int \psi_\theta^2(\boldsymbol{r}) \, \mathrm{d}\, \boldsymbol{r}}.$$
(22)

If we now define the probability density function (PDF) $p(\boldsymbol{r}) = \frac{\psi^2(\boldsymbol{r})}{\int \psi^2(\boldsymbol{r}) \, \mathrm{d}\, \boldsymbol{r}}$, we can rewrite Equation 22 as

$$E_0 \leq \int p(\boldsymbol{r}) \frac{\boldsymbol{H} \psi_\theta(\boldsymbol{r})}{\psi_\theta(\boldsymbol{r})} \, \mathrm{d}\, \boldsymbol{r} = \int p(\boldsymbol{r}) E_L(\boldsymbol{r}) \, \mathrm{d}\, \boldsymbol{r} = \mathbb{E}_{p(\boldsymbol{r})} \left[ E_L(\boldsymbol{r}) \right]$$
(23)

where the right-hand side is the so-called VMC energy. By taking gradients of the VMC energy to the parameters $\theta$ of the trial wave function $\psi_\theta$, we can optimize the parameters to minimize the energy. These gradients can be computed as

$$\nabla_\theta = \mathbb{E}_{p(\boldsymbol{r})} \left[ (E_L(\boldsymbol{r}) - \mathbb{E}_{p(\boldsymbol{r})} \left[ E_L(\boldsymbol{r}) \right]) \nabla_\theta \log \psi_\theta(\boldsymbol{r}) \right].$$
(24)

By approximating the expectation values via Monte Carlo sampling, we can compute the gradients via Equation 24 and optimize the parameters via gradient descent (Ceperley et al., 1977).

## G   Setup

This section details the setup of our experiments.

As Jastrow factors, we use an MLP on the averaged electronic coordinates:

$$J(\boldsymbol{r}) = \mathrm{MLP} \left( \sum_{i=1}^{N} [r_i - R_m, \|r_i - R_m\|]_{m=1}^{M} \right).$$
(25)

We picked this specific formulation as it provides a similar computational cost to the odd function defined in Section 3 as it does not explicitly consider the individual electronic coordinates. We use this Jastrow as a standalone Jastrow or the symmetric function in Equation 5 and 6.

As standard in the field, we pretrain the FermiNet on a Hartree-Fock wave function and then optimize the odd function with the pretrained FermiNet within the VMC framework (Pfau et al., 2020).

We implement everything in JAX (Bradbury et al., 2018). To compute the laplacian from Equation 21, we use the forward laplacian algorithm from Li et al. (2023) implemented by Gao et al. (2023).

We use the hyperparameters listed in Table 2 for our experiments.

## H   Additional Experiments

In addition to the experiments in the main body, we present additional empirical data here. As an alternative to the KFAC optimizer, we present results with prodigy (Mishchenko & Defazio, 2023), a learning-rate-free version of Adam (Kingma & Ba, 2014), in Figure 5. However, the conclusion remains the same: The odd functions improve energies on small structures and worsen

| Hyperparameter | Value |
|---|---|
| **CASSCF** | |
| Basis set | 6-311G |
| **FermiNet** | |
| Determinants | 8 |
| Single electron features | 256 |
| Pairwise features | 32 |
| Layers | 4 |
| Activation | SiLU |
| **Odd functions** | |
| Layers | 5 |
| Hidden units | 256 |
| $\alpha$ | -2 |
| **Jastrow** | |
| Layers | 3 |
| Hidden units | 256 |
| **Optimization** | |
| Optimizer | KFAC |
| Learning rate | $\frac{0.05}{1+\frac{t}{1000}}$ |
| Damping | 0.001 |
| Batch size | 2048 |
| MCMC steps | 20 |

Table 2: Hyperparameters used in the experiments if not otherwise specified.

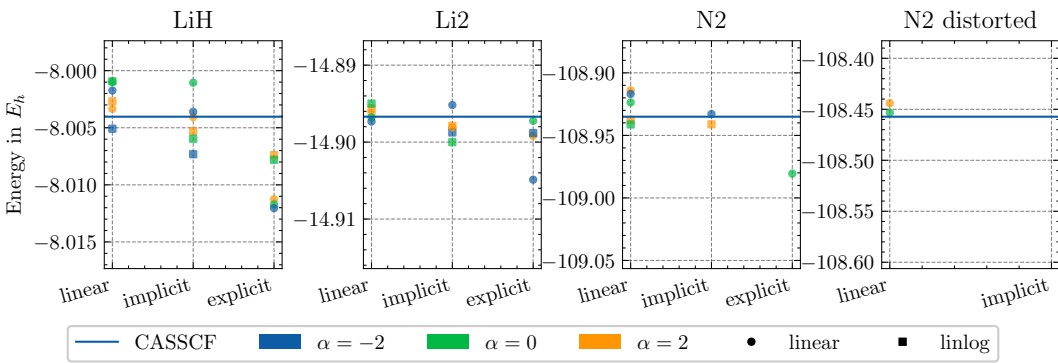

Figure 5: Final energies of CASSCF+odd optimized with Prodigy. Missing entries were numerically unstable and encountered NaNs during training.

them on large ones. Further, we present CASSCF experiments with Jastrow factors in Figure 6. Here, we observe that including a Jastrow factor closes the gap between the linear and non-linear combinations. Finally, we present CASSCF experiments in float64 precision in Figure 7 to show that the numerical instabilities are not due to numerical precision.

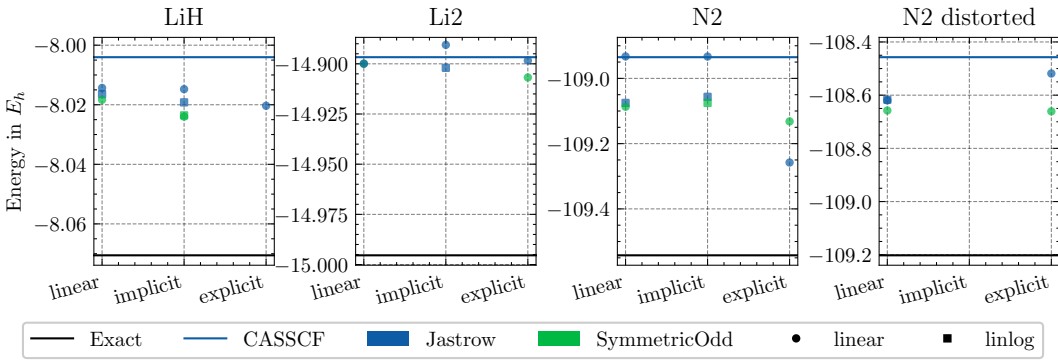

Figure 6: The final energies of CASSCF+odd optimized with KFAC or Prodigy. Missing entries encountered NaNs during training. Colors indicate the use of symmetric functions: Traditional Jastrow factor, and 'SymmetricOdd' implies including the symmetric Jastrow factor as in Equation 5, and 6. $\alpha = -2$.

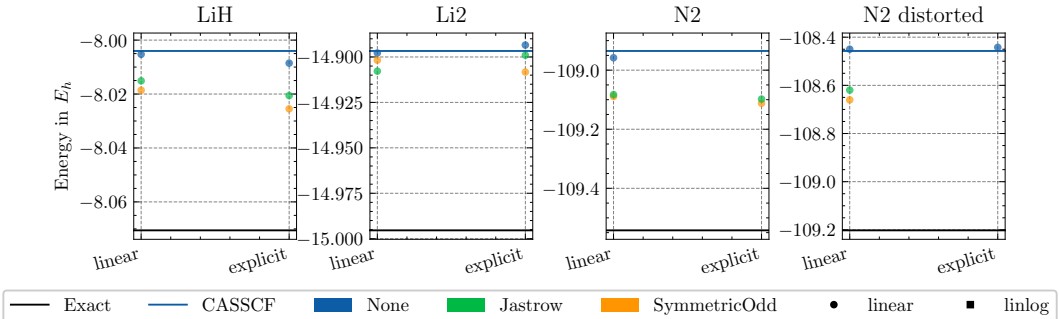

Figure 7: Final energies of CASSCF+odd optimized with Prodigy. Training in float64. $\alpha = -2$.

