# OpenReview forum: "On Representing Electronic Wave Functions with Sign Equivariant Neural Networks"
_ICLR.cc/2024/Workshop/AI4DiffEqtnsInSci — AI4DiffEqtnsInSci @ ICLR 2024 Poster_

### Official Review · Reviewer_jYk2 · 2024-02-18
**Strong paper which is focused more on the application than on machine learning**

**Rating:** 8
**Confidence:** 3

**Review:**

## Summary
The authors present a method to approximate the electronic ground-state wave functions in molecules. To this end, related work has mostly proposed to approximate individual wave functions within the Slater-determinant using neural networks. These determinants are then combined linearly and multiplied with a Jastrow factor. In contrast, this paper suggests to replace this linear combination with a neural network which is constructed to guarantee the anti-symmetric property of the wave function. The authors provide proof that the network structure guarantees the anti-symmetric property. They go on to show, that for every function that can be represented by their neural network there is a Jastrow-Slater-Wavefunction that is identical to it wherever (phi(r)^T x) is non-zero. In empirical experiments they do not find a clear advantage of their method.

## Suitability for this workshop
I think the results are interesting to the workshop as the authors build a specialized neural network to ensure certain properties of the solution.

## Comments
While I would have wished for more details on the neural networks for this venue, I think that given the page limit, the selection of information is comprehensible. I think that the paper is clearly written. It was sometimes difficult to follow for me as the paper requires some background in quantum mechanics that I lack. However, given the page limit I do not think a longer introduction to this topic is feasible. I think the discussion of the results is good, only the final sentence of the abstract might be a slight over-claim because it is difficult to claim it being impossible from it not showing in one experiment.

##  Justification of Rating
I think this is a very good work as it presents empirical experiments that are well motivated and further gives good theoretical insights why these empirical experiments turned out the way they did. The only downside in my opinion is that the focus is much more on quantum mechanics and less on learning representations. Nevertheless, I think this paper should clearly be accepted.

---

### Official Review · Reviewer_Et4K · 2024-02-26

**Rating:** 6
**Confidence:** 2

**Review:**

**Summary**
The paper investigates the utility of sign equivariant neural networks that combine determinants *non-linearly* for representing electronic wave functions. This approach is different from existing neural-network-based approximations to the wave function that rely on *linear* combinations of determinants (so-called Slater-Jastrow wave function).

Specifically, the paper contains two main contributions:
1. A theoretical result, showing that any sign equivariant non-linear combination of determinants can be equivalently represented using the classical "linear" Slater-Jastrow form.
2. An empirical evaluation of two classes of wave functions based on sign equivariant neural networks. The evaluation shows that the sign equivariant architectures do not provide a consistent advantage over the traditional methods based on the Slater-Jastrow form.

**Strengths/weaknesses**

The paper studies an important ML problem, provides a clear presentation of the results, and examines an interesting question overlooked by prior work. The findings of this paper may be important in shaping the direction of future work in this ML area.

My only criticism of the paper is that the conclusion in the abstract ("sign equivariant functions are unsuitable for representing electronic wave functions") might be too strong given the evidence presented in the paper.

The theoretical result shows that the non-linear component can be in theory "absorbed" into the permutation invariant Jastrow factor $J(r)$, while the $\det \Phi(r)$ terms are used to capture the fermionic antisymmetry to permutations. The experimental evaluation studies two specific sign equivariant NN architectures. Neither of these statements excludes the possibility that different architectures for $J(r)$ or the sign equivariant aggregation function $f(x, y)$ would lead to a different conclusion.  As a somewhat far-fetched example, the strong conclusion presented in the paper is akin to concluding that neural networks are unsuitable for learning functions of images by 1/ invoking the universal approximation property of MLPs and 2/ showing their poor results on some image-related ML tasks. I would recommend to add more nuance to the statement, taking the limitations of the experimental setup into account.

**Minor suggestions:**
- Typos in Definition 2, should instead be "the group acts on $\mathcal{Y}$ as $\pi y \mapsto \operatorname{sign}(\pi) y$"
- The description of "Linear-logarithmic domain" is Section 3 feels out of place. The statement about samples drawn from $\phi^2$ refers to the experiments that come much later. Also, previous paragraphs use $x$ to denote the vector of determinants, but in the definition of $\operatorname{linglog}$, $x$ is a scalar
- Potential typo: $f^{(t)}$ instead of $f^{(T)}$ before equation 5.

Disclaimer: I have limited experience in ML for quantum chemistry, so please feel free to downweight my feedback in favor of reviewers who are experts in this domain.

---

### Meta-Review · Area_Chair_26Xa · 2024-03-01

**Recommendation:** Accept (Poster)

**Metareview:**

Thanks to reviewers for their careful review and nice suggestions and questions. Both reviewers seem to agree with the acceptance oft his paper. I also went through their points and the paper, and recommend authors to address them in their final revision.

---

### Decision · Program_Chairs · 2024-03-02

Accept (Poster)